# The small eye phenotype in the EPIC-Norfolk eye study: prevalence and visual impairment in microphthalmos and nanophthalmos

Alexander C Day,[1] Anthony P Khawaja,[2] Tunde Peto,[1] Shabina Hayat,[2] Robert Luben,[2] David C Broadway,[3] Kay-Tee Khaw,[2] Paul J Foster[1]

[1]The NIHR Biomedical Research Centre at Moorfields Eye Hospital NHS Foundation Trust and UCL Institute of Ophthalmology, London, UK
[2]Department of Public Health and Primary Care, Institute of Public Health, University of Cambridge School of Clinical Medicine, Cambridge, UK
[3]Department of Ophthalmology, Norfolk and Norwich University Hospital NHS Foundation Trust and University of East Anglia, Norwich, UK

Correspondence to
Professor Paul Foster; p. foster@ucl.ac.uk

## ABSTRACT

**Objective:** To describe the prevalence and phenotypic characteristics of small eyes in the European Prospective Investigation of Cancer (EPIC)-Norfolk Eye Study.

**Design:** Community cross-sectional study.

**Setting:** East England population (Norwich, Norfolk and surrounding area).

**Participants:** 8033 participants aged 48–92 years old from the EPIC-Norfolk Eye Study, Norfolk, UK with axial length measurements. Participants underwent a standardised ocular examination including visual acuity (LogMAR), ocular biometry, non-contact tonometry, autorefraction and fundal photography. A small eye phenotype was defined as a participant with one or both eyes with axial length of <21 mm.

**Outcome measures:** Prevalence of small eyes, proportion with visual impairment, demographic and biometric factors.

**Results:** Ninety-six participants (1.20%, 95% CI 0.98% to 1.46%) had an eye with axial length less than 21 mm, of which 74 (77%) were women. Prevalence values for shorter axial lengths were <20 mm: 0.27% (0.18% to 0.41%); <19 mm: 0.17% (0.11% to 0.29%); <18 mm: 0.14% (0.08% to 0.25%). Two participants (2.1%) had low vision (presenting visual acuity >0.48 LogMAR) and one participant was blind (>1.3 LogMAR). The prevalence of unilateral visual impairment was higher in participants with a small eye. Multiple logistic regression modelling showed presence of a small eye to be significantly associated with shorter height, lower body mass index, higher systolic blood pressure and lower intraocular pressure.

**Conclusions:** The prevalence of people with small eyes is higher than previously thought. While small eyes were more common in women, this appears to be related to shorter height and lower body mass index. Participants with small eyes were more likely to be blind or to have unilateral visual impairment.

## INTRODUCTION

The small eye phenotype ranges from anophthalmos to nanophthalmos and

### ARTICLE SUMMARY

**Article focus**
- The European Prospective Investigation of Cancer-Norfolk Eye Study is part of a European population-based cohort study, with participants now aged 48–92 years old.
- This paper describes the prevalence of small eyes, proportion with visual impairment, and associated demographic and biometric factors.

**Key messages**
- Ninety-six participants of 8033 (1.20%, 95% CI 0.98% to 1.46%) had an eye with axial length less than 21 mm, of which 74 (77%) were women.
- People with small eyes appear more likely to be blind or have unilateral visual impairment. Presence of a small eye is associated with shorter height, lower body mass index, higher systolic blood pressure and lower intraocular pressure.
- There are no standardised definitions for microphthalmos or nanophthalmos.

**Strengths and limitations of this study**
- Large population-based study sample.
- The included population sample may have healthy volunteer bias.
- The identified associations are cross-sectional rather than longitudinal.

microphthalmos. The latter two conditions are typically considered to be synonymous[1] and are subdivided into simplex[2] and complex[3] depending on the presence of other associated ocular or systemic abnormalities. There is minimal adult data on the prevalence of this phenotype with estimated birth prevalences for microphthalmos being 0.002–0.017%;[4] and 0.009% for microphthalmos in China from mass screening programmes.[5] Data from a hospital cohort suggest that patients with simple microphthalmos comprise between 0.05% and 0.11%

ophthalmic patients.[6] There is great heterogeneity in the definition of nanophthalmos and microphthalmos which complicates interpretation of previous studies,[2 7–12] with a definition by axial length (AL) <21 mm being the most inclusive.[10–12] Nanophthalmos/microphthalmos is associated with angle closure glaucoma;[1 13] and also with significant visual morbidity. In a recent series of nanophthalmic individuals from a Melanese population almost half had either unilateral or bilateral visual impairment.[11] There is a paucity of data for comparison. In view of this, we report data on the prevalence and characteristics of small eyes in British adults in the European Prospective Investigation of Cancer (EPIC)-Norfolk Eye Study, and review the definitions used for microphthalmos and nanophthalmos.

## METHOD

EPIC is a pan-European study that started in 1989 with the primary aim of investigating the relationship between diet and cancer risk.[14] The aims of the EPIC-Norfolk cohort were subsequently broadened to include additional endpoints and exposures such as lifestyle and other environmental factors.[15] The EPIC-Norfolk cohort was recruited between 1993 and 1997 and comprised 25 639 predominantly white European participants aged 40–79 years. The third health examination was carried out between 2006 and 2011 with the objective of investigating various physical, cognitive and ocular characteristics of participants then aged 48–92 years.[16] The third health examination was reviewed and approved by the East Norfolk and Waverney NHS Research Governance Committee (2005EC07L) and the Norfolk Research Ethics Committee (05/Q0101/191) and was performed in accordance with the principles of the Declaration of Helsinki. All participants gave written, informed consent.

All EPIC-Norfolk Eye Study participants underwent a detailed health examination performed by trained nurses following standard operating procedures. Ocular biometry was measured by non-contact partial coherence interferometry using the Zeiss IOLMaster Optical Biometer (IOLMaster, Carl Zeiss Meditech Ltd, Welwyn Garden City, UK). Five measurements of AL as well as anterior chamber depth (ACD, defined as corneal epithelium to anterior crystalline lens surface) and three measurements of central keratometry were made to allow calculation of mean values. Refractive error was measured using an autorefractor (Model 500, Humphrey Instruments, San Leandro, California, USA). Three intraocular pressure (IOP) measurements were made for each participant using the non-contact Ocular Response Analyzer (ORA, Reichert Inc, Depew, New York, USA) and the mean Goldmann correlated IOP (IOPg) calculated. Visual acuity was measured under standardised conditions at 4 m using participants' normal method of distance vision correction and recorded on the LogMAR scale. Fundal photographs were taken of both eyes using a TRC-NW65 non-

mydriatic retinal camera (Topcon Corporation, Tokyo, Japan) with Nikon D80 camera (Nikon Corporation, Tokyo, Japan). A masked, expert grader from the Moorfields Grading Centre measured vertical cup–disc ratio (VCDR). Systolic and diastolic blood pressures (BPs) were taken from the right arm with the participant seated for 5 min. A stadiometer was used to record participant height to the nearest 0.1 cm and weight was measured to the nearest 0.1 kg using a body composition analyser (Tanita model TBF 300 s, Chasmors Ltd, London, UK). Self-reported data on education, occupation, alcohol intake and smoking status were recorded by questionnaire.

A small eye was defined by an AL of <21 mm in at least one eye in keeping with the broadest previously accepted definition for microphthalmos/nanophtahlamos[10–12] and being equivalent to 2SD below the population mean value.[17] All investigations were performed on both eyes of each participant and the data from the eye with lower AL used for analyses at the participant level, with the exception of visual impairment classification where data from both were used. Visual impairment was defined by the presenting vision in accordance with the International Classification of Diseases Update and Revision 2006[18] and the WHO, which formally comprises categories 1–5 with categories 3–5 being blindness. To allow comparison with previous publications we defined blindness as a presenting visual acuity ≥1.3 logMAR in the better eye and low vision as a presenting visual acuity of >0.48 in the better eye (ie, combination of moderate and severe visual impairment categories). Unilateral visual impairment was defined by using the eye with worse presenting visual acuity.

Statistical analysis was performed using SPSS V.20. Testing of normality was performed by the Kolmogorow-Smirnov method. Comparisons between participants with and without previous lens extraction were performed using the independent samples t test or Mann-Whitney U test. Logistic regression was used to identify factors associated with presence of a small eye and Fisher's exact test to compare presence of visual impairment with the presence of a small eye.

## RESULTS

Partial coherence interferometry data were available on 15 881 eyes of 8033 participants, of which 4442 participants were women (55.3%). Case numbers and overall prevalence values for small eyes stratified by AL value are shown in table 1 and figure 1. Of the 8033 participants with AL data, visual acuity measurements were available on 8016 (99.8%).

Of the 96 participants, 20 were pseudophakic in both eyes, 6 were pseudophakic in one eye, 1 was aphakic in both eyes (congenital cataracts and nystagmus) and 1 aphakic in one eye and pseudophakic in the other. Defined by smallest eye, 26 participants had undergone previous lens extraction. Fourteen participants (15%)

**Table 1** Number of participants/eyes and overall prevalence values (with 95% CIs) by axial length (mm)

| Axial length (mm) | Analysis by participant | | Analysis by eyes | |
| --- | --- | --- | --- | --- |
| | Number | Prevalence (95% CI) | Number | Prevalence (95% CI) |
| <21.00 | 96 | 1.195% (0.980 to 1.457) | 132 | 0.831% (0.702 to 0.985) |
| <20.50 | 47 | 0.585% (0.441 to 0.777) | 57 | 0.359% (0.277 to 0.465 |
| <20.00 | 22 | 0.274% (0.182 to 0.414) | 24 | 0.151% (0.102 to 0.225) |
| <19.00 | 14 | 0.174% (0.105 to 0.292) | 14 | 0.088% (0.053 to 0.148) |
| <18.00 | 11 | 0.137% (0.077 to 0.245) | 11 | 0.069% (0.039 to 0.124) |
| <17.00 | 4 | 0.050% (0.020 to 0.127) | 4 | 0.025% (0.010 to 0.065) |
| <16.00 | 1 | 0.012% (0.003 to 0.069) | 1 | 0.006% (0.002 to 0.035) |
| <15.00 | 1 | 0.012% (0.003 to 0.069) | 1 | 0.006% (0.002 to 0.035) |

had a history of amblyopia or previous squint surgery. Seven participants (7%) had a history of previous laser iridotomy or surgical iridectomy. Table 2 shows the demographic and biometric characteristics of those with AL <21 mm.

Analysis of the difference in AL between eyes showed a bimodal distribution (figure 2) with 19 participants (20%) comprising the second peak with a mean AL difference of 5.63 mm (SD 0.97) compared with 77 participants in the first peak with mean AL difference of 0.45 mm (SD 0.39).

Both univariable and multiple variable regression analyses investigating ocular biometric parameters in phakic eyes showed small eyes were associated with shallower ACD, steeper corneal keratometry and higher spherical equivalent (all p<0.001, table 3). Separate analyses were performed for other, non-ocular biometric parameters. For these, univariable logistic regression analyses showed female sex (OR 2.75, p<0.001), height (per 10 cm, OR 0.46, p<0.001), weight (per 10 kg, OR 0.60, p<0.001), body mass index (BMI, OR 0.68, p=0.005) and systolic BP (per 10 mm Hg, OR 1.11, p=0.029) were associated with the presence of a small eye. Multiple variable logistic regression models showed shorter height, lower BMI,

higher systolic BP and lower IOP to be independent predictors of a small eye (table 3).

Optic disc grading was possible on both eyes of 61/96 (64%) participants and at least one eye of 82/96 (85%) participants (right eyes: 12 missing, 9 ungradable; left eyes: 14 missing, 14 ungradable). Three participants (3/61, 4.9%) had VCDR asymmetry of 0.2 or more, and one additional participant had an optic disc consistent with glaucoma (localised absence of neural rim, one eye only), giving an overall prevalence of 4/61 (6.6%, 95% CI 2.6% to 15.7%) for glaucomatous optic neuropathy. No eye had a VCDR of ≥0.6. Five of 96 (5.2%) participants gave a diagnosis of 'glaucoma' in their medical history, of these only one had a diagnosis consistent with their optic disc photographs. Three participants had one optic disc with disc drusen. There were no cases of macular hypoplasia, macular schisis, coloboma or any other retinal abnormality associated with nanophthalmos.

Visual acuity data were available for all 96 participants and values are shown in table 2. One participant (1%) was classified as blind by the WHO definition (visual acuity of less than 1.3 logMAR) and 2/96 (2.1%) had any degree of visual impairment. Using a definition of visual impairment of >0.30 logMAR in the better eye to allow comparison with previous visual impairment studies, the prevalence was 5/96 (5.2%). The prevalence of blindness was significantly higher in EPIC-Norfolk participants with at least one eye of AL <21 mm compared with those without, while the overall prevalence of low vision was similar (table 4). Unilateral visual impairment by all definitions was more common in EPIC-Norfolk participants with at least one small eye compared with those without (p≤0.001, table 4).

## DISCUSSION

There are minimal data describing the prevalence or characteristics of small eyes. In the EPIC-Norfolk Eye Study, the prevalence of a participant with an eye of AL of <21 mm was 1.20%, and 0.27% for those with an eye of AL <20 mm. Relative to existing data with estimated birth prevalences for microphthalmos between 0.002% and 0.017%;[4][5] and the prevalence of simple microphthalmos in hospital ophthalmic patients being

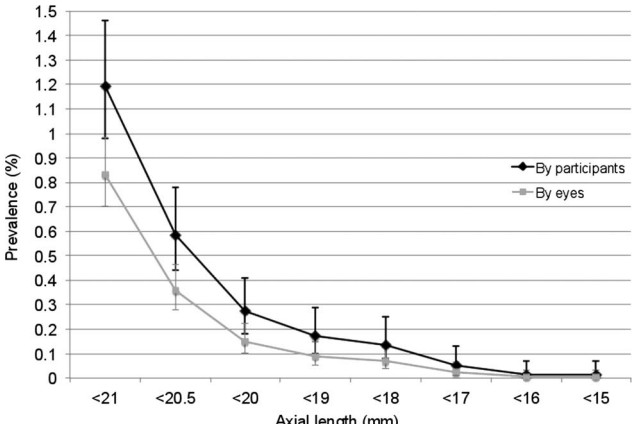

**Figure 1** Graph showing the prevalence of an axial length less than the value shown at the participant level (defined by eye with shortest axial length), and at the eye level.

**Table 2** Demographic and biometric data presented as mean values with (standard deviation and range min: max value (range for all participants only)), with [median values, IQR] shown for AL & ACD only

|  | All, axial length <21 mm | Phakic | Previous lens extraction | p Value |
|---|---|---|---|---|
| Number | 96 | 70 | 26 | – |
| Age (years) | 69.0 (8.8, 50.9 to 89.2) | 66.3 (7.5) | 76.5 (7.6) | <0.001 |
| Sex | 22M/74F | 13M/57F | 9M/17F | 0.11 |
| AL (mm) | 20.05 (1.26, 14.27 to 20.98) | 20.45 (0.85) | 18.96 (1.55) | <0.001 |
|  | [20.53, 0.80] | [20.61, 0.48] | [18.91, 2.85] |  |
| ACD (mm) | 2.94 (0.69) | 2.67 (0.44) | 3.75 (0.71) | <0.001 |
|  | [2.75, 0.78] | [2.62, 0.44] | [3.98, 0.92] |  |
| Mean K (D) | 45.24 (1.62, 41.71 to 51.19) | 45.45 (1.65) | 44.64 (1.41) | 0.044 |
| SE (D) | +3.63 (2.94, −5.50 to +8.38) | +5.04 (1.84) | −0.15 (1.71) | <0.001 |
| Anisometropia, (D) | 1.13 (1.23, 0.00 to 6.76) | 1.20 (1.27) | 0.94 (1.09) | 0.37 |
| V/A (logMAR) | 0.31 (0.47, −0.20 to 1.68) | 0.37 (0.53) | 0.16 (0.24) | 0.061 |
| LogMAR difference between eyes | 0.31 (0.44, 0.00 to 1.82) | 0.38 (0.49) | 0.12 (0.19) | 0.012 |
| IOP (mm Hg) | 15.7 (3.8) | 15.6 (3.9) | 16.0 (3.3) | 0.63 |

Comparisons with p values are between phakic and those with previous lens extraction.
ACD, anterior chamber depth; AL, axial length; IOP, intraocular pressure.

between 0.05% and 0.11%;[6] small eyes are more common than previously reported.

Nanophthalmos is traditionally associated with a high prevalence of angle closure glaucoma[1 13] and both nanophthalmos and primary angle closure glaucoma have similar ocular phenotypes including a short AL, shallow anterior chamber, hyperopia, small radius of corneal curvature and a thick crystalline lens. In this study, the prevalence of glaucomatous optic neuropathy was estimated to be 6.6% in small eyes based on CDR, CDR asymmetry or rim abnormalities consistent with glaucoma. Othman et al[19] reported that 12/22 (55%) nanophthalmic individuals had occludable anterior chamber angles or glaucoma. In the case series by Tay et al[11] of 17 individuals with nanophthalmos, no data on glaucoma prevalence are reported. For comparison, the prevalence of glaucoma (open angle and closed angle combined) has been estimated to be 2.4% in European populations.[20] We did not calculate VCDR percentiles for the overall EPIC cohort as per the International Society Geographical & Epidemiological Ophthalmology definition of glaucoma;[21] however, no small eye in our series had a VCDR of ≥0.60. Crowston et al[22] reported optic disc size adjusted VCDR percentiles from the Blue Mountains Eye Study and showed that for small discs (1.2 mm vertical diameter) in non-glaucomatous eyes, the 97.5th centile for VCDR was 0.60 (99th centile: 0.62) while corresponding values for large optic discs (1.9 mm diameter) were 0.75 and 0.83, respectively. Thus our prevalence of 6.6% should be considered as a minimum prevalence estimate for glaucoma in small eyes, and is likely to include those at highest risk for visual impairment over their lifetime.[21]

Review of recent studies reporting on nanophthalmos/microphthalmos shows great heterogeneity in the definitions used, with cases defined primarily by short AL with for example, values of <21,[10–12] <20.9,[2] <20.5,[7] <20,[8] <18,[2] or <17 mm.[9] The original description of nanophthalmos (or pure microphthalmos) by Duke-Elder[1] is an eye 'reduced in volume without the presence of other gross congenital abnormalities,' 'typical dimensions are 16–18.5 mm sagittal,' 'hyperopia is the rule' and 'the anterior chamber is typically shallow.' The partial relaxation of the definition to its currently accepted form (of at least an AL of <21 mm) is likely due to the rarity of the condition. If an abnormally short eye is defined based on the lower 2SD and 3SD limits of mean population ALs, then the calculated limits are approximately 21 and 20 mm, respectively.[16] In a previous study by our group investigating complications in small eyes (<21 mm) undergoing phacoemulsification and lens implantation, only AL and the presence of abnormal IOP remained significant predictors of any complication in multiple variable regression analysis.[12] Complications were 15 times more likely in cases

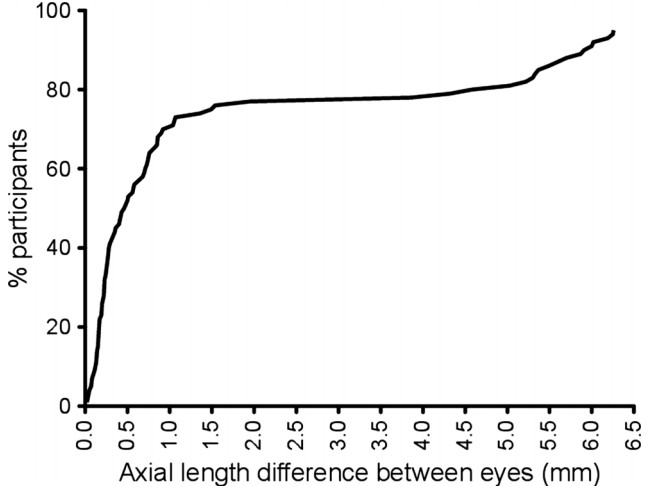

**Figure 2** Cumulative frequency distribution of axial length difference (asymmetry) between eyes for each participant. Note the bimodal distribution.

**Table 3**  Univariable and multiple variable logistic regression analyses of factors associated with small eyes

|  | OR | 95% CI | p Value |
|---|---|---|---|
| *(A) Ocular biometric parameters: phakic participants* | | | |
| Univariable regression | | | |
| Anterior chamber depth (per 1 mm) | 0.06 | 0.03 to 0.12 | <0.001 |
| Mean keratometry (per 1D) | 2.16 | 1.82 to 2.57 | <0.001 |
| Spherical equivalent (per 1D) | 2.67 | 2.35 to 3.03 | <0.001 |
| Multiple variable regression | | | |
| Anterior chamber depth (per 1 mm) | 0.02 | 0.01 to 0.08 | <0.001 |
| Mean keratometry (per 1D) | 5.97 | 3.98 to 8.98 | <0.001 |
| Spherical equivalent (per 1D) | 5.89 | 4.16 to 8.31 | <0.001 |
| *(B) Other parameters: all participants* | | | |
| Univariable regression | | | |
| Age (per decade) | 1.06 | 0.82 to 1.36 | 0.67 |
| Female sex | 2.75 | 1.70 to 4.43 | <0.001 |
| Height (per 10 cm) | 0.46 | 0.36 to 0.58 | <0.001 |
| Weight (per 10 kg) | 0.60 | 0.51 to 0.72 | <0.001 |
| BMI (per 5 kg/m$^2$) | 0.68 | 0.52 to 0.89 | 0.005 |
| Social class | | | |
| Professional | Ref | | |
| Managerial/technical | 0.81 | 0.39 to 1.69 | 0.57 |
| Skilled non-manual | 0.91 | 0.40 to 2.09 | 0.82 |
| Skilled manual | 0.95 | 0.43 to 2.10 | 0.90 |
| Partly-skilled | 1.06 | 0.44 to 2.53 | 0.90 |
| Unskilled | 1.76 | 0.53 to 5.77 | 0.35 |
| Education level | | | |
| Less than O level | Ref | | |
| O level | 1.31 | 0.69 to 2.50 | 0.41 |
| A level | 0.94 | 0.57 to 1.55 | 0.81 |
| Degree | 0.93 | 0.50 to 1.76 | 0.83 |
| Systolic blood pressure (per 10 mm Hg) | 1.11 | 1.01 to 1.23 | 0.029 |
| Diastolic blood pressure (per 10 mm Hg) | 0.97 | 0.78 to 1.20 | 0.78 |
| Self-reported alcohol intake | | | |
| No intake | Ref | | |
| <7 units/week | 0.81 | 0.48 to 1.37 | 0.43 |
| ≥7<14 units/week | 0.61 | 0.33 to 1.13 | 0.12 |
| ≥14<21 units/week | 0.61 | 0.28 to 1.33 | 0.22 |
| ≥21 units/week | 0.70 | 0.38 to 1.28 | 0.25 |
| Smoking status | | | |
| Never | Ref | | |
| Ever | 0.85 | 0.56 to 1.27 | 0.41 |
| Intraocular pressure (mm Hg) | 0.95 | 0.90 to 1.01 | 0.09 |
| Multiple variable regression | | | |
| Age (per decade) | 0.89 | 0.68 to 1.17 | 0.40 |
| Female sex | 0.91 | 0.47 to 1.77 | 0.77 |
| Height (per 10 cm) | 0.42 | 0.29 to 0.59 | <0.001 |
| BMI (per 5 kg/m$^2$) | 0.69 | 0.53 to 0.90 | 0.006 |
| Systolic blood pressure (per 10 mm Hg) | 1.11 | 1.01 to 1.22 | 0.030 |
| Intraocular pressure (mm Hg) | 0.93 | 0.88 to 0.99 | 0.030 |

Ref: reference category. For the multiple variable regression models (either A or B), only parameters reaching statistical significance in the respective univariable analysis were included, and only those in the final model shown.
BMI, body mass index.

with AL of <20 mm (compared with those 20–21 mm, p≤0.001). The differential complication rate supports the previous recommendation by Weiss *et al*[2] that microphthalmos and nanophthalmos should be considered as two separate phenotypes, based on AL. Based on the above it would appear reasonable to classify small eyes into microphthalmos (<21 mm) and nanophthalmos (<20 mm), respectively.

We found 1% of participants with small eyes were blind and 2% had low vision. When compared with all EPIC-Norfolk participants, blindness appears to be more common in those with a small eye (p=0.036, although case numbers were very low), but low vision was not (p=0.36). Unilateral visual impairment (defined by the worse seeing eye) was more common by all definitions (p≤0.001). When compared with data from population studies, the

**Table 4** Percentages of bilateral and unilateral visual impairment in participants with one or both eyes with axial length <21 mm (n=96) compared with all EPIC-Norfolk participants with no eye of axial length <21 mm (n=7920) by Fisher's exact test

| LogMAR | Snellen equivalent | Classification | EPIC-Norfolk participants without small eyes (n=7920 total) | | EPIC-Norfolk participants with small eyes (n=96 total) | | p Value |
|---|---|---|---|---|---|---|---|
| | | | n | Prevalence (95% CI) | n | Prevalence (95% CI) | |
| >1.30 better eye | <3/60; 20/400 | WHO blindness | 2 | 0.03% (0.00, 0.06) | 1 | 1.0% (0.0, 3.1) | 0.036 |
| >0.48 better eye | <6/18; 20/60 | Blindness and visual impairment; 'low vision' | 45 | 0.6% (0.4, 0.7) | 2 | 2.1% (0.0, 5.0) | 0.11 |
| >0.22 better eye | <6/10; 20/32 | UK driving standard | 422 | 5.3% (4.8, 5.8) | 7 | 7.29% (2.0, 12.56) | 0.36 |
| >0.30 better eye | <6/12; 20/40 | Previous visual impairment studies, American driving standard | 259 | 3.3% (2.9, 3.7) | 5 | 5.2% (0.7, 9.7) | 0.25 |
| >1.0 worse eye | <6/60; 20/200 | Unilateral visual impairment | 120 | 1.5% (1.3, 1.8) | 11 | 11.5% (5.0, 18.0) | <0.001 |
| >0.48 worse eye | <6/18; 20/60 | Unilateral visual impairment | 470 | 5.9% (5.4, 6.5) | 24 | 25.0% (16.2, 33.8) | <0.001 |
| >0.30 worse eye | <6/12; 20/40 | Unilateral visual impairment | 1341 | 16.9% (16.1, 17.8) | 29 | 30.2% (20.9, 39.6) | 0.001 |

Bilateral visual impairment is defined as both eyes with a visual acuity less than the respective value and unilateral visual impairment as one eye with a visual acuity less than the respective value.
EPIC, European Prospective Investigation of Cancer.

prevalence of visual impairment in EPIC-Norfolk participants is low overall, and values in those with small eyes are again low or similar. In the Blue Mountains Eye Study,[23] 4.6% had a visual acuity of 6/12 (20/40) or less in the better eye and 14.4% had a visual acuity of 6/12 or less in their worse eye, whereas in our cohort of small eyes the equivalent percentages were 5.2% and 30.2%. In the Salisbury Eye Study, 9% of participants aged 75–84 years old had a visual acuity of <6/12 in their better eye;[24] while in the MRC study in Britain this value was 15% for those 75–84 years old.[25] There is minimal data on visual impairment in nanophthalmic individuals, with a recent study in a Melanesia population[11] (definition: AL usually <21 mm in at least one eye) reporting 5/17 (29%) had bilateral visual impairment and 9/17 (53%) had unilateral visual impairment (defined as <6/12 (20/40) Snellen in the better eye).

We found a marked bimodal distribution in AL difference between eyes in individuals with small eyes, with 20% individuals having >3.5 mm AL asymmetry. A bimodal distribution in AL difference has not previously been described, with Weiss *et al*[2] reporting a difference of only 0.4 mm or less in a series of 21 patients with simple microphthalmos.

Our study has a number of limitations; these being primarily the absence of lens thickness and scleral thickness data to further characterise participants with small eyes. Additionally participants were not examined on a slit-lamp, for example, gonioscopy to determine the presence of an occludable anterior chamber angle (and therefore to determine if the glaucomatous optic neuropathy in our five cases were in the presence of an open or closed anterior chamber angle). Our prevalence value for glaucoma was based on glaucomatous optic neuropathy only rather than glaucomatous optic neuropathy and visual field defect.[21] Comparisons of visual acuities were only performed in EPIC-Norfolk participants in whom ALs were measurable, and consequently this may have excluded those with visual impairment or blindness where AL could not have been measured optically (ie, underestimating prevalence values). Additionally, those with visual impairment may have self-selected not to participate in the EPIC-Norfolk Eye Study, thus again underestimating case numbers.

In summary, the small eye phenotype was more common than previously reported, and our study provides prevalence values in British adults. There are no standardised definitions for microphthalmos or nanophthalmos; however, based on current evidence, subdivision by AL of <21 mm for microphthalmos and <20 mm for nanophthalmos appears reasonable. People with small eyes appear more likely to be blind or have unilateral visual impairment. The estimated prevalence of glaucomatous optic neuropathy in our cohort appeared to be lower than expected and warrants further investigation.

**Contributors** ACD performed the data analysis and drafted the manuscript. APK and TP contributed to the data analysis. SH and DCB contributed to the conception and design of the study. RL contributed to the design of the study

and to the data acquisition and management. K-TK and PJF contributed to the conception and design of the study, and to the data interpretation. All authors read and critically revised the manuscript and approved the final manuscript.

**Funding** Supported by grant G0401527 from the Medical Research Council, UK and grant 262 from Research into Ageing, UK. APK is funded by a Wellcome Trust Clinical Research Fellowship. PJF was also supported by the Richard Desmond Charitable Trust (via Fight for Sight, grant 1956). ACD, TP & PJF were supported by the National Institute for Health Research (NIHR) Biomedical Research Centre based at Moorfields Eye Hospital NHS Foundation Trust and UCL Institute of Ophthalmology. The views expressed are those of the author(s) and not necessarily those of the NHS, the NIHR or the Department of Health.

**Competing interests** None.

**Ethics approval** Norwich Local Research Ethics Committee (05/Q0101/191) and East Norfolk & Waveney NHS Research Governance Committee (2005EC07L).

**Provenance and peer review** Not commissioned; externally peer reviewed.

**Data sharing statement** The data sharing and preservation strategy in EPIC-Norfolk is in accordance with the Wellcome Trust data management and sharing policy. Full details about the study including contact information are on the website http://www.epic-norfolk.org.uk. Investigators wishing to work with EPIC data contact the EPIC management group through the website, letter, phone or fax and proposals have to fulfil a number of criteria including that the work is within the bounds of consent given by participants and has been ethically reviewed and approved; there is no serious risk to the viability of continuing the cohort study, for example, through offence to the participants from use of the data supplied; the science of the proposal has been satisfactorily peer reviewed and the proposal does not duplicate work already being done. Access to data for collaborators is provided through password protected website access. The large numbers of collaborators EPIC-Norfolk has locally, nationally and internationally (>300), as evidenced by collaborative publications, demonstrate the commitment to maximising the value of the study.

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
