## [Reviewer comments · BMJ Open]

Some articles will have been accepted based in part or entirely on reviews undertaken for other BMJ Group journals. These will be reproduced where possible.

ARTICLE DETAILS

TITLE (PROVISIONAL)	The Small Eye Phenotype in the EPIC-Norfolk Eye Study: Prevalence and Visual Impairment in Microphthalmos and Nanophthalmos.
AUTHORS	Day, Alex; Khawaja, Anthony; Peto, Tunde; Hayat, Shabina; Luben, Robert; Broadway, David; Khaw, KayTee; Foster, Paul

VERSION 1 - REVIEW

REVIEWER	Mr Hiten G. Sheth Consultant Ophthalmologist Stoke Mandeville Hospital Aylesbury Buckinghamshire HP21 8AL No competing interests.
REVIEW RETURNED	03-Jun-2013

GENERAL COMMENTS	1. Possible typo in introduction p5 line 10 'simplex'.2. The editor should note that some data may overlap with the smaller cohort published previously in Foster PJ, Broadway DC, Hayat S et al. Refractive error, axial length and anterior chamber depth of the eye in British adults: the EPIC-Norfolk Eye Study. Br J Ophthalmol 2010;94:827-30. The authors may wish to comment and provide clarification on any overlap in data sets (directly to the editor).3. Their attempt to classify / define microphthalmos vs nanophthalmos is helpful and will encourage more precise use of these terms in future by other authors.4. The finding of not infrequent axial length asymmetry is novel and intriguing. - This paper is authored by a respected collaborative group of practising ophthalmologists also involved in epidemiological research. It follows on seamlessly from the EPIC-Norfolk Eye Study rationale, methods and visual impairment paper published in BMJ Open March 2013. - The results highlight data on the ocular anatomy of what constitutes a largely Caucasian white population in this part of the UK. To date there has been a paucity of small eye phenotype data in Caucasian populations, as the authors allude to. The data is also of interest as it gives the prevalence of the small eye phenotype in the 'normal' population rather than in the context of severe visual
--

	impairment from early life. - The data, particularly prevalence rates, are useful to practising ophthalmologists who can now convey to patients some perspective on how common or uncommon their particular biometry is. This will be of value in discussing risk for those undergoing lens extraction surgery where there is an increased chance of intra-operative complications such as capsule rupture and post-operative complications such as aqueous misdirection. - The authors identify the possibility of recruitment bias towards healthy individuals, but this is unlikely to have significantly impacted on the objectives of this particular paper which focussed more on structural aspects rather than ocular symptoms or disease. They also state the limitations relating to lack of lens thickness and scleral thickness data.
--	--

REVIEWER	Francesco M Quaranta-Leoni MD Chief, Oftalmoplastica Roma Via Francesco Siacci 39 - 00197 Roma - Italy No conflicts of interest to disclose
REVIEW RETURNED	11-Jun-2013

- The reviewer completed the checklist but made no further comments.